# Can Gait Characteristics Be Represented by Physical Activity Measured with Wrist-Worn Accelerometers?

**DOI:** 10.3390/s23208542

**Published:** 2023-10-18

**Authors:** Wenyi Lin, Fikret Isik Karahanoglu, Dimitrios Psaltos, Lukas Adamowicz, Mar Santamaria, Xuemei Cai, Charmaine Demanuele, Junrui Di

**Affiliations:** Pfizer Inc., Cambridge, MA 02139, USAcharmaine.demanuele@pfizer.com (C.D.); junrui.di@pfizer.com (J.D.)

**Keywords:** wearable devices, accelerometry, gait, physical activity, device placement, penalized regressions, partial least square, joint and individual variation explained, multilevel model

## Abstract

Wearable accelerometers allow for continuous monitoring of function and behaviors in the participant’s naturalistic environment. Devices are typically worn in different body locations depending on the concept of interest and endpoint under investigation. The lumbar and wrist are commonly used locations: devices placed at the lumbar region enable the derivation of spatio-temporal characteristics of gait, while wrist-worn devices provide measurements of overall physical activity (PA). Deploying multiple devices in clinical trial settings leads to higher patient burden negatively impacting compliance and data quality and increases the operational complexity of the trial. In this work, we evaluated the joint information shared by features derived from the lumbar and wrist devices to assess whether gait characteristics can be adequately represented by PA measured with wrist-worn devices. Data collected at the Pfizer Innovation Research (PfIRe) Lab were used as a real data example, which had around 7 days of continuous at-home data from wrist- and lumbar-worn devices (GENEActiv) obtained from a group of healthy participants. The relationship between wrist- and lumbar-derived features was estimated using multiple statistical methods, including penalized regression, principal component regression, partial least square regression, and joint and individual variation explained (JIVE). By considering multilevel models, both between- and within-subject effects were taken into account. This work demonstrated that selected gait features, which are typically measured with lumbar-worn devices, can be represented by PA features measured with wrist-worn devices, which provides preliminary evidence to reduce the number of devices needed in clinical trials and to increase patients’ comfort. Moreover, the statistical methods used in this work provided an analytic framework to compare repeated measures collected from multiple data modalities.

## 1. Introduction

In recent years, due to advancements in sensor technologies, there has been a rapid growth in the use of wearable devices to obtain high-resolution and objective measures of physical function and behavior in participants’ free-living environments, for both public health research and clinical trials [1]. Wearable devices provide opportunities to characterize human health beyond conventional snapshots of in-lab measurements by capturing a holistic picture of patients’ daily life, including sitting, walking, climbing, cycling, sleep, etc. [2].

Among wearable devices, accelerometers, typically a key component of most wearable activity trackers and smartphones, are commonly used to track 24-h motor activity such as walking, climbing, cycling, and sleep [1,3,4]. Walking gait characteristics are important functional measurements and have been shown to be associated with aging-related functional and cognitive decline, falls, and mortality and can be used as clinical endpoints for multiple conditions [5,6,7,8,9,10]. Research on physical activity (PA) focuses on summarizing overall activity volumes and categorizing the waking period into different activity intensities, such as sedentary, light, moderate, and vigorous [1,11], and studying the distributions and transitions of different intensities. A wealth of evidence shows that time spent in sedentary behaviors is associated with adverse health outcomes and increased time spent in sedentary behaviors increases the severity and complications associated with those health outcomes [12,13,14]. In recent years, accelerometry-measured PA has been gradually accepted to be used as primary or secondary endpoint for pivotal clinical studies [15,16].

The location of the accelerometer on the body can impact the prediction of activity types [3]. Lumbar-worn and wrist-worn devices are two commonly used types of devices. There has been extensive literature demonstrating that the lumbar location can provide a reliable estimation of temporal and spatial gait parameters [17,18,19,20,21]. Multiple algorithms were developed to extract gait features based on signals from lumbar-worn accelerometers [17,22,23]. On the other hand, wrist-worn devices are typically accurate in quantifying PA in general and number of steps per day [24]. The underlying assumption of employing accelerometers to quantify physical activity hinges on establishing a correlation between the acceleration signal and energy expenditure, where this correlation is established through controlled laboratory-based activity tasks, during which oxygen consumption is measured. Numerous studies have been dedicated to constructing and validating associations for difference cohorts. Below, we provide a selection of noteworthy studies that have contributed significantly to this field of research [3,25,26,27]. More importantly, due to the high comfort and user acceptability of a wrist-worn device (i.e., nowadays people are already used to wearing smart watches), it has been used in a number of large cohort studies to objectively quantify daily PA for multiple days [28,29,30,31].

In drug development, it is of interest to select reliable and sensitive endpoints collected from the most appropriate body location. For example, to evaluate the treatment effect for Duchenne muscular dystrophy, stride velocity 95th percentile is considered a regulatory acceptable clinical endpoint [32], which can be measured by a lumbar-worn accelerometer [33]. From an operational point of view, it is important to consider patient burdens and preferences when conducting a long-term clinical trial since this impacts compliance and data completeness. Having to wear a device in the lumbar region for weeks or months may not be ideal for patients. On the other hand, wrist-worn devices have been shown to offer better compliance when compared with lumbar-worn devices due to ease and comfort of wear resulting in lower patient burden [34,35], and therefore may be preferred by patients [36]. Choosing the right sensor location based on cohort and study design to obtain the endpoints of interest while maintaining operational simplicity is critical for successful deployment in clinical trials.

To evaluate whether it is possible to avoid having to deploy both lumbar- and wrist-worn devices in a clinical study, we need to explore the association between gait and PA features measured at the two locations. Studies have demonstrated that specific gait features such as gait speed are associated with PA [1,31,37]. Those studies typically only focus on a small number of PA or gait features and are hard to be generalized to the two domains of gait and activity with a full spectrum of features from each domain. Moreover, results were based on accelerometry-measured PA and lab-measured gait speed. There is still a lack of systematic evaluation of the association and joint information between a comprehensive list of gait and PA features in free-living environments. It calls for statistical innovation to explore their correlation while incorporating the totality of PA and gait and also the repeated-measurement nature of the data collection. And, it has not been established if gait features measured by lumbar-worn devices can be represented by PA features measured by wrist-worn devices.

The purpose of this paper is to fill the gap by evaluating if gait features measured by lumbar-worn devices can be represented by PA features measured by wrist-worn devices. In the remainder of the paper, we first review multiple statistical models to evaluate the univariate and multivariate association between measurements from the gait domain and the PA domain. These methods range from simple feature-to-feature correlations, including linear regression analysis and principal component regression, to more complex multivariate dimension reduction methods, such as partial least square (PLS) [38] and joint and individual variation explained (JIVE) [39]. Since the features from lumbar and wrist devices are generally collected for multiple days, we evaluated the association between gait and PA based on both summary statistics (one-level) across seven days and repeated measures recorded daily (multilevel). We then demonstrate how these methods can be leveraged in a real data application where both wrist-worn and lumbar-worn devices were deployed with the same participants simultaneously.

## 2. Evaluate the Association and Joint Information from the Two Placements

A large number of algorithms have been proposed in the literature to evaluate the association or the joint information of data from different domains. In this section, we review and summarize commonly used methods in three parts: (1) feature-on-domain models, (2) domain-on-domain models, and (3) multilevel models. Figure 1 summarizes statistical models implemented for one-level and multilevel data. In this work, one-level methods refer to models applied to features averaged over days. Multilevel models are extended from one-level models, which incorporate repeated daily measurements of each feature for each participant. Feature-on-domain models are implemented only on one-level data, including pairwise correlation analysis, penalized linear regression [40], and principal component regression (PCR) [41]. Domain-on-domain models involve multivariate and multidomain analysis for both one-level and multilevel data, such as partial least square (PLS) regression [38] and joint and individual variation explained (JIVE) [39].

### 2.1. Notation

We use the *P* dimension vector $x_{ij}=\left(x_{ij1},\ldots ,x_{ijP}\right)$ and *Q* dimensional vector $y_{ij}=\left(y_{ij1},\ldots ,y_{ijQ}\right)$ to denote the *P* features in the PA domain and *Q* features in the gait domain, respectively, for subject i,i=1,…,n at day $j,j=1,\ldots ,n_{i}$. Let $x_{p}={\left\{x_{ijp}\right\}}_{i=1,\ldots ,n,j=1,\ldots ,n_{i}}$ be the *p*th feature vector in PA domain and $y_{q}={\left\{y_{ijq}\right\}}_{i=1,\ldots ,n,j=1,\ldots ,n_{i}}$ be the *q*th feature vector in PA domain. These vectors are stacked to form the PA feature domain $X=(x_{1},\ldots ,x_{P})$ and the gait feature domain $Y=(y_{1},\ldots ,y_{Q})$. We use $N=\sum_{i}n_{i}$ to denote the total number of observations. In our proposed one-level models (Section 2.2 and Section 2.3), the mean values across days, i.e., $x_{i.p}=\frac{1}{n_{i}}\sum_{j=1}^{n_{i}}x_{ijp}$ and $y_{i.q}=\frac{1}{n_{i}}\sum_{j=1}^{n_{i}}y_{ijq}$, are incorporated. In these models, $x_{ip}$ and $y_{iq}$ are used for notation simplicity. We use (.)’ to denote the transpose of a vector or a matrix.

### 2.2. Feature-on-Domain Models

First, correlation and regression analysis are implemented to explore each feature individually from both gait and PA domains and its association with the other domain. The Pearson correlation coefficient is a measure of linear correlation between two sets of data, with the form of the covariance of the two variables divided by the product of their standard deviations. For each pair of features $(x_{p},y_{q}),p=1,\ldots ,P,q=1,\ldots ,Q$, from wrist- and lumbar-worn devices, respectively, we compute their corresponding correlation coefficients $\rho_{x_{p},y_{q}}$.

In addition to pairwise correlation evaluation, linear multiple regression models are included to further explore the linear associations between two domains, via regressing each univariate variable from the gait domain on the PA domain. Multicollinearity within the PA feature domain can be addressed with enforced sparsity via Lasso regressions, and the model selection is based on a five-fold cross-validation procedure incorporating ’one-standard-error’ rule, that is, choosing the model with the fewest coefficients that is less than one standard error away from the sub-model with the lowest error. Therefore, the penalized linear regression is utilized to explore the linear association between a single feature from the one measurement domain with multiple features from the other measurement domain, extending the one-on-one correlation analysis.

Another common approach to address multicollinearity when analyzing multiple regression data is principal component regression (PCR) [41]. Specifically, principal component analysis (PCA) was applied on the PA domain to transform it into an orthogonal predictor space with a lower dimension. In our implementation, the number of principal components (PCs) is selected based on finding the “elbow” point in the scree plot. PCR is implemented to find the features in the PA domain which explain the most variance, as well as to find the features in the gait domain which have the highest association with the PCs.

### 2.3. Domain-on-Domain Models

We expand our modeling strategy to quantify the associations between the gait and PA domains using multivariate models. These models, referred to as domain-on-domain models, provide a more comprehensive analysis of the relationship between gait and PA features.

#### 2.3.1. Partial Least Squares Regression

The partial least squares (PLS) regression [38] is a technique that combines features from PCA and multiple linear regression, which seeks to find the multidimensional direction in the PA domain X that explains the maximum multidimensional variance direction in the gait domain Y. In essence, the goal of PLS regression is to find a pair of unit vectors μ and ν so that the expression
(1)$$\begin{array}{c} \mu^{\prime }X^{\prime }Y\nu , \end{array}$$
is maximized. The process is implemented with R package *pls*. The resulting relation between gait and PA domains can be explained with the correlation and loading plots. Compared with the PCR model introduced in Section 2.2, the components obtained from the PLS regression are based on covariance from both X and Y domains so that they can explain the mutual association.

#### 2.3.2. Joint and Individual Variation Explained

JIVE has been proposed to deal with scenarios where different sources or views of the data are simultaneously available for the same set of samples [39]. JIVE decomposes the original multiblock data into a sum of three components: a low-rank approximation capturing joint variation of the domains, low-rank approximations capturing individual variation in each domain, and residual noise. The imposed rank and orthogonality constraints can be considered as extensions of PCA. Let $Z=\left(\begin{array}{c} X^{\prime }\\ Y^{\prime } \end{array}\right)$, represent the combined feature matrix with dimension (P+Q)×N. The JIVE can be formulated as follows
(2)$$\begin{array}{cc} X^{\prime } & =J_{1}+A_{1}+\epsilon_{1}\\ Y^{\prime } & =J_{2}+A_{2}+\epsilon_{2}, \end{array}$$
where $\epsilon_{i}$ is the error matrix of independent entries with $E(\epsilon_{i})=0$, for i=1,2. $J=\left(\begin{array}{c} J_{1}\\ J_{2} \end{array}\right)$ is the joint structure matrix with rank *r* and $A_{l}$ denotes the individual structure with respect to X and Y with lower dimensional ranks $r_{1}$ and $r_{2}$, respectively. Row orthogonality is enforced between both joint and individual components. JIVE was fitted using the R package *r.jive*.

### 2.4. Multilevel Models

The above mentioned feature-on-domain and domain-on-domain models are all considered one level because they are implemented on the mean feature values recorded at subject level. They can be further extended to multilevel models which include repeated measures for each subject, since both lumbar-worn and wrist-worn devices are normally worn continuously for multiple days.

#### 2.4.1. Multilevel Partial Least Square Regression

The multilevel PLS regression [42] borrows the idea from the two-way ANOVA model, which decomposes the original process into the sum of three parts: overall mean, between-subject effect, and within-subject effect; i.e., for the observed feature vector $x_{ij}$ (or $y_{ij}$), it can be decomposed into
(3)$$\begin{array}{c} x_{ij}=\mu +\left(x_{i.}-\mu \right)+\left(x_{ij}-x_{i.}\right), \end{array}$$
where μ is the mean term with the form $\frac{1}{\sum n_{i}}\sum_{i,j}x_{ij}$. $\left(x_{ij}-x_{i.}\right)$ is the within-subject matrix with $x_{i.}=\frac{1}{n_{i}}\sum_{j}x_{ij}$, and $\left(x_{i.}-\mu \right)$ represents the between-subject matrix. The same format of transformation can be implemented on $y_{ij}$. For both within- and between-subject matrices, the proposed PLS regression in Section 2.3.1 is performed. Therefore, both within-subject and between-subject relations between gait and PA domains can be explained via the multilevel PLS regression model.

#### 2.4.2. Multilevel JIVE

To incorporate repeated measurements from multidomain data, Di et al. (2019) proposed multilevel JIVE [43] that combines multilevel PCA (MPCA) [44] and JIVE.

Multilevel JIVE utilizes a two-step procedure. First, a two-way ANOVA like decomposition is performed via MPCA to separate the joint variation into random between- and within-subject effects ($u_{i}$,$w_{ij}$, respectively) with lower dimensional presentation, as shown in
(4)$$\begin{array}{c} z_{ij}=\mu +u_{i}+w_{ij}, \end{array}$$
where μ is the overall mean, and $z_{ij}=\left(x_{ij},y_{ij}\right)$ is the P+Q dimensional vector for all features. The number of retained principal components is based on a pre-specified percentage of explained variation.

As the second step, JIVE is applied to both the between- and within-subject effects ($u_{i}$,$w_{ij}$, respectively) to extract the joint and individual structure, as shown in
(5)$$\begin{array}{cc} u_{i}^{\prime } & =J_{u_{i}}+A_{u_{i}}+\epsilon_{u_{i}}\\ w_{ij}^{\prime } & =J_{w_{ij}}+A_{w_{ij}}+\epsilon_{w_{ij}}. \end{array}$$

Therefore, the proposed multilevel JIVE is able to explain the joint and individual effects at both subject and day levels.

## 3. Real Data Application: The STRYDE Study

### 3.1. Subjects and Instrumentation

The Sensors to Record Your Daily Exercise (STRYDE) study was conducted at the Pfizer Innovation Research (PfIRe) Lab in Cambridge, Massachusetts. The study recruited 65 healthy participants in total, with 33 of them in the younger (18–40) cohort (age = 29.2±4.6 years, 17 females) and 32 in the older (65–85) cohort (age = 72.3±5.8 years, 16 females). Details of the study were previously published [33]. In general, the study was designed to have participants perform the same walking tasks during two in-clinic visits and be continuously monitored in their free-living environment for 7–14 days.

Specfically for this study, we only focused on the gait and PA measurements in the free-living enviroments. Continuous assessment of functional activities at home was collected using two Activinsights GENEActiv devices [45], with one placed at the lumbar, and the other on the non-dominant wrist. Devices recorded tri-axial accelerometer data (range: ±8 g, sampling rate: 50 Hz, resolution: 12 bits/3.9 mg) and were attached to the body using straps. A schematic representation of a participant wearing the device and participating in the gait analysis is provided in Figure 2. Data were stored locally on the device and downloaded for offline processing following the return of the device to the study site.

### 3.2. Data Processing

Raw acceleration signals from lumbar-worn GENEActiv devices were used to derive gait features. Gait features were generated using the implementation in the open-source SciKit Digital Health (SKDH) (https://github.com/PfizerRD/scikit-digital-health (accessed on 30 August 2023)) Python library [22]. The gait algorithm first detects gait bouts from continuous free-living data using a gradient-boosted tree classifier and then applies a continuous wavelet transformation [46] and inverted pendulum model [20] to derive temporal and spatial gait features for steps in each gait bout. We followed previous practice to first remove bouts that lasted <10 s or >3000 s, then take the median of gait features across all steps within each bout, and eventually take the mean (and/or 95th percentile) across all gait bouts per day [33]. Step counts and bout lengths were summed up across all bouts within each day. Eleven gait features were derived, namely total step counts per day, total bout length per day, gait speed, cadence, stance time, swing time, stride duration, double support, single limb support, stride length, and 95th percentile of gait speed.

Raw signals from wrist-worn GENEActiv devices were used to derive PA features. The Microsoft Excel Macros accompanying the GENEActiv device were used to generate the PA features [47]. The algorithm generated minute-by-minute epochs in each 24-h period and classified them into sleep, wear, bed, no wear, wear time (excluding bed time), sedentary activity, light activity, moderate activity, and vigorous activity time and eventually generated the relevant activity features. The activity features were extracted based on summarizing and thresholding the signal vector magnitude (SVMg). A total of thirteen PA features were presented in this study, including: total duration and numbers of sedentary, light, moderate, and vigorous PA bouts, total daily sleep time, and non-thresholded features, such as mean and 95th percentile of SVMg and mean activity level during the most active 6/15/60-min-window per day.

For both devices, we followed the convention to keep only valid days for each participant, defined as having 10 h or more of wearing time of wrist-worn devices [48,49]. Table 1 provided a list of gait and PA features along with their corresponding abbreviations. Summary metrics of these features for all participants, as well as for younger and older cohorts, were included in Table 2. On average, participants in the study had 8.78 (±1.86) valid days. To deal with the large discrepancy in the scales between different gait and PA features, all features were centered and scaled. For the one-level model, for each gait and PA feature, averages across days were taken. Sensitivity analysis showed that the model performance was similar if the median or quantile value across days was used.

### 3.3. Results—One-Level Models

The linear correlation between the gait and PA features was analyzed using pairwise Pearson correlation coefficients and is displayed in Figure 3. The within-domain correlations for gait features revealed patterns that potentially group gait features into three clusters, i.e., spatial features related to gait quality (including stride duration, cadence, double support, single limb support, stance, and swing times), spatial and spatio-temporal features (stride length, gait speed, 95th percentile of gait speed total steps), and features related to gait quantity (total steps and total bout length). The results also indicated that gait quantity measures, i.e., total steps and total bout length, have a stronger correlation with PA features. In PA domain, features related to peak activities during the day were grouped, such as maximum activity in the most active 6/15/60 window, moderate/vigorous activity, and mean/95th percentile of SVMg, while sedentary to light activities formed another group.

We applied linear multiple regression models with a Lasso penalty to evaluate the relationship between the two domains. Figure 4 displays the coefficients from the Lasso regression for the selected predictors in each regression model. The results showed that gait quantity features, i.e., total steps and total bout length, are positively associated with time spent in moderate activity and mean SVMg.

PCA was applied to the PA features and the first four components explained over 90% of total variation in the PA domain. Figure 5 displays the estimated eigenvectors of the top two PCs. Specifically, the first PC (PC1) was heavily loaded on mean SVMg, 95% percentile SVMg, and moderate activity, while the second PC (PC2) was heavily loaded on light activity and number of light and sedentary activity bouts. We fitted the PCR with the top four PCs as the predictors and each gait feature as the response. The estimated coefficients are displayed in Figure 6, which further supported the strong association between total bout length and total steps per day, and the activity features, especially in the first and fourth PCs.

In the domain-on-domain models, we aimed to uncover the association between two domains including all features. Figure 7 displays the contribution of gait and PA features on the top two components estimated from PLS. It first showed that gait features, including total bout length, total steps per day, 95th percentile of gait speed, mean gait speed, and stride length heavily contributed to the first component in the same direction, while double support, single limb support, stance, and swing times contributed to the first component in the opposite direction. This suggests that within all gait features, there are two latent clusters. The first cluster includes spatial and spatio-temporal features (e.g., gait speed and stride length) and gait quantity features (e.g., total steps per day and total bout length). And another cluster consists of gait quality features, such as double support and stride duration. Moreover, total steps per day, total bout length, mean and 95% percentile gait speed, and stride length contributed to the first component in a similar fashion to almost all PA features (i.e., negative loadings on the first component) except for sleep and sedentary time. Interestingly, PA features that summarized peak activity during the day (e.g., maximum activity in the most active 6/15/60 window, vigorous activity, and 95th percentile of SVMg) and the gait quantity features contributed to both components in a similar fashion, which suggests that they are highly correlated.

We further applied JIVE to explore the joint variation shared by the two domains. We identified a rank-1 representation of the joint structure and rank-2 representations of the gait and PA individual structure. Table 3 provides the contribution of both gait and PA features to the variation explained by the joint and individual components. Overall, the joint component explained 30% of the total variation of the combined gait and PA domain and the individual components contributed slightly more, which explained 56% of the total variation. In total, the joint and individual components explained over 85% of the total variation in two domains. Fifty eight percent of the joint variation was contributed by gait and the remaining forty-two percent by PA. These results suggest that the domains shared a great amount of joint information, with the gait domain contributing to the joint information slightly more than the PA domain. On the contrary, the PA domain explained more variation in individual components than the gait domain.

### 3.4. Results—Multilevel Models

In the multilevel models, the day-to-day variations in PA and gait features were taken into account. The results from the multilevel PLS regression demonstrated that the between- and within-subject effects were dominated by total steps per day and total bout length in the gait domain. Figure 8 shows the estimated first component for both between and within-subject effects. Particularly, total bout length and total steps per day had the highest contributions from the gait domain, followed by moderate time, mean, and 95th percentile of SVMg from the activity domain, within both between- and within-subject effects. These findings strengthen the conclusion that total steps per day and total bout length were closely related to the PA domain and played a significant role in the association between gait and PA domains.

Furthermore, the two-step multilevel JIVE was implemented. In the first step, MPCA was applied to all the features to separate between-subject and within-subject effects. The estimated between-subject and within-subject effects both explained around 45% of the total variation, which demonstrated the importance of not overlooking the within-subject (day-to-day) variation (Table 4). In the second step, JIVE was applied on between-subject effects and within-subject effects with main results shown in the second part of Table 4, which indicated that around 50% joint variation was shared between the two domains for both between- and within-subject effects. This demonstrated the equal importance of two domains in explaining the joint variance. The individual components explained 42% and 49% of the between- and within-subject effects, respectively. Therefore, for both between- and within-subject effects, over 95% of the total variation was covered by the joint and individual components. Figure 9 and Figure 10 illustrate the cross-correlation between features from the lower-rank representation of the between-subject effect $u_{i}$ and within-subject effect $w_{ij}$ and the extracted JIVE scores from the corresponding multilevel JIVE models. Similar grouping patterns of features were found in between-subject and within-subject effects. Gait quantity features (i.e., total bout length, total steps) and gait speed contributed to the joint component (first column) in the same direction as most of features from the PA domain (except for sedentary and sleep features). Furthermore, these gait features exhibited the highest correlation with the joint component, indicating that they are more likely to be represented by the PA features. Regarding the gait individual component in within-subject effects (as shown in the second column in Figure 10), it was observed that PA features related with more vigorous activities, such as maximum activity in the most active 6/15/60 window and 95th percentile of SVMg, contributed to this component in the same direction as gait speed, stride length, and cadence.

### 3.5. Comparison between One-/Multilevel PLS/JIVE

We then compared the results from the one-level and multilevel models. The estimated lower dimension components from the one- and the multilevel PLS and JIVE are shown in Figure 11. The first component in each model, which explains the largest amount of variance, was always highly correlated with the first components from the other models, indicating that the results are consistent across models. In particular, the strongest correlations were observed between the first principal components of the two one-level models (PLS.PC1 vs. JIVE.JT.PC1) and between the between-subject and within-subject effects of the multilevel models (PLS.B.PC1 vs. PLS.W.PC1 and JIVE.B.JT.PC1 vs. JIVE.W.JT.PC1).

## 4. Discussion

In this paper, we systematically investigated the association between gait features measured with lumbar-worn accelerometers and PA features measured with wrist-worn accelerometers, which, to the best of our knowledge, is the first study of its kind. We applied multiple univariate and multivariate statistical models to data collected from the STRYDE study to evaluate the association based on the average feature values across days and repeated measurements across days. Our results showed that gait and PA are interrelated and share joint information. Specifically, gait quantity measures, such as total steps and total bout length, and gait speed show a higher correlation with PA features when compared to other gait features. As a result, they are more likely to be represented by a combination of PA features, including those summarizing peak activity during the day, such as maximum activity within the most active 6-, 15-, and 60-min windows, vigorous activity, and the 95th percentile of SVMg. This result has been corroborated by multiple univariate and multivariate methods, such as multilevel JIVE and multilevel PLS.

We implemented multiple statistical models, including both univariate and multivariate methods which can be categorized into feature-on-domain analyses and domain-on-domain analyses. Specifically, the feature-on-domain analyses aim to reveal the association between each individual gait feature with all PA features. The domain-on-domain analyses aim to reveal the overall covariance and shared variation between the two domains of gait and PA. We specifically highlighted the use of JIVE, which has been previously used to explore the joint association among different physiological domains of PA, sleep, and circadian rhythms [1]. JIVE can be further applied in multimodal or multidomain data analyses. The day-to-day variation in continuous measurements from wearable devices is a well-known fact [50] and is associated with multiple health-related phenomena, such as social jetlag [51] and weekend warrior [52]. Therefore, in this work, we also assessed the importance of using multilevel analyses to take into account the within-subject correlation due to the repeated daily measurements, which are often ignored by only considering average values across days. Both one-level and multilevel models yielded consistent results, demonstrating that gait quantity features can be represented by PA features. The multilevel models provided further insights—it showed that both between-subject and within-subject effects explain 50% of the total variation in the data, which indicates that individual characteristics and day-to-day variability play important roles in understanding the associations between the gait and activity domains.

Previous research has indeed established connections between specific gait characteristics and PA features. For instance, Wanigatunga et al. identified associations between faster gait speed and several PA metrics, including increased activity counts, daily active minutes, and reduced activity fragmentation [31]. Similarly, Schrack et al. also observed a correlation between gait speed and activity fragmentation [37]. However, these studies primarily focused on gait speed as a singular gait characteristic and relied on lab-measured gait speed rather than accelerometer-based gait characterization. Moreover, Dawe et al. [53] discovered that three gait and balance measures were independently associated with total daily physical activity, assessed through total activity counts. However, it is important to highlight that the “gait and balance measures” in this research encompassed a combination of activities such as walking, sit-to-stand, and turning, extending beyond a pure gait characterization. These measurements were derived from a structured mobility testing protocol conducted in a controlled laboratory environment, which differs from the accelerometer-based gait characterization in free-living environment used in our study. Nevertheless, our study reinforces the consistent conclusion that gait speed is indeed correlated with various PA features. Importantly, we achieved this through a comprehensive analytical framework that encompasses a wide range of gait and PA characteristics. This approach not only reaffirms the relationship between gait and PA but also underscores the potential clinical significance of these correlations, particularly in understanding patients’ prognoses. Future research endeavors should explore these connections further, delving into longitudinal trends to provide valuable insights into the evolving relationship between PA and gait in clinical contexts.

The association between gait features and PA features identified in this work provides preliminary evidence that a single wrist-worn device can be deployed in a clinical trial. Specifically, as our results suggest, total gait quantity and gait speed exhibit a strong association with PA features, which indicates that they can be analytically represented by PA features. Furthermore, in scenarios where the primary interest lies in assessing total gait quantity or gait speed, PA features hold potential as surrogate endpoints. From an operational point of view, deploying a single wrist-worn, watch-like device will increase compliance and reduce data loss due to difficulty in wearing the devices. This will be a good step further toward decentralized clinical trials and more patient-centric clinical trials in general [11,54]. However, for therapeutic areas where specific spatial and spatio-temporal features are of clinical interest, such as stride duration, step symmetry, or gait variability, it may still be preferable to use the more fit-for-purpose devices, such as the lumbar-worn devices [55].

There has been recent research attempting to directly quantify walking gaits from wrist-worn accelerometers. However, it is worth noting that challenges persist when utilizing a single wrist-worn accelerometer for comprehensive and reliable gait characterization. For instance, Trost et al. [56] employed logistic regressions to classify various activity types, including walking, using both wrist and hip-worn devices. While their walking detection achieved reasonable accuracy, extracting precise gait metrics from wrist-accelerometry signals during walking bouts remains non-trivial. Similarly, Sokas et al. [57] detected 6-min windows of fast walking in daily activities using wrist-worn accelerometers. They counted steps in these windows and used piece-wise linear models with covariates like age and height to estimate walking distance which potentially can be hard to scale. While valuable for estimating distance in free-living 6-min walk task, this approach did not delve into detailed spatial and spatio-temporal gait characteristics. Brand [58] explored machine learning and deep learning techniques to classify gait based on free-living data. However, their summarization of walking bouts was limited to total daily walking time and a few time and frequency domain features, lacking the comprehensive gait characterization which can be typically obtained from lumbar-worn devices. Soltani et al. [59] proposed a personalized machine learning method for gait speed estimation using accelerometers but relied on Global Navigation Satellite System’s calibration and correction, limiting its generalizability. Chan et al. [60] employed support vector machine (SVM) models for gait bout classification and speed estimation but faced challenges in characterizing a wide range of gait characteristics. Additionally, the non-physical nature of the SVM model may not fully capture the underlying mechanics of walking. Therefore, the need for further investigation into wrist-worn-accelerometer-based gait characterization is evident, and lumbar-worn accelerometers remain the preferred choice for reliable and comprehensive gait characterization, surpassing the mere detection of walking bouts. Future research may leverage this framework to examine cross-domain associations between wrist-based activity, lumbar-based gait, and wrist-based gait once more reliable wrist-based gait estimation becomes available.

A significant contribution of this work is to provide a comprehensive analytic framework to study the inter-correlation and association between multiple data modalities with repeated measurements. Using this framework, we comprehensively evaluate the relationship between PA and gait, taking into account the repeated measurement design and a wide spectrum of PA and gait characteristics. This approach distinguishes our research from previous studies with similar objectives. We have implemented our methods within a hierarchical fashion, making it more accessible for researchers with less statistical background in this field. As demonstrated in this work, while one could consider a subset of the introduced methods to assess the association between features from different domains, our comprehensive implementation has provided new insights and interpretations from various perspectives. Additionally, cross-checking between methods has been performed to validate the results obtained from them. In this paper, even though the two modalities are gait and PA features from lumbar- and wrist-worn devices, respectively, such an analytic framework can be generalized to other comparisons involving repeated measurements to explore the associations, interactions, and shared information across different data modalities, domains, and/or time points.

We acknowledge that the biggest limitation of this work is the relatively small sample size. One important future work is to validate our findings in a separate study with a larger sample size and with both lumbar- and wrist-worn devices deployed, which may be challenging. Another direction to pursue validation is to pool multiple studies with small sample sizes which utilize both device locations. If the studies use different brands of devices, a device-agnostic data processing pipeline, such as Scikit Digital Health (SKDH) [22], could be used to generate gait and PA features across studies. With lager sample sizes, it is possible to evaluate the age effect on the association between gait and PA features, which may provide additional clinical insights. Furthermore, it is important to note that the STRYDE study primarily relied on data collected from a sample of healthy adult participants. While the findings from this study offer valuable insights, it is essential to exercise caution when extrapolating these results to individuals with pathological conditions. To establish a broader applicability of the conclusions drawn from the STRYDE study, additional research involving participants who have specific medical conditions or pathology may be necessary. In those cases, our proposed analytic framework can be directly applied.

## Figures and Tables

**Figure 1 sensors-23-08542-f001:**
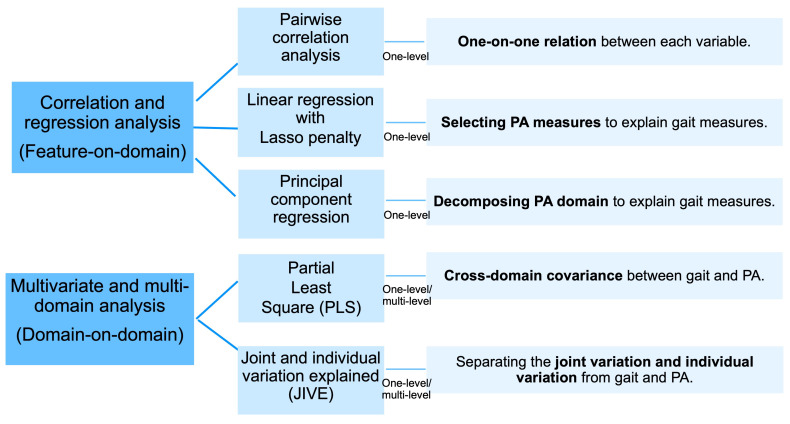
Overview of the analytical framework implemented in this paper for comparing multivariate, multilevel, and multidomain data.

**Figure 2 sensors-23-08542-f002:**
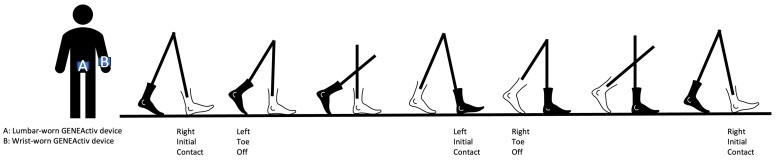
A schematic representation of devices placement of a participant (**left**) and gait characteristics (**right**).

**Figure 3 sensors-23-08542-f003:**
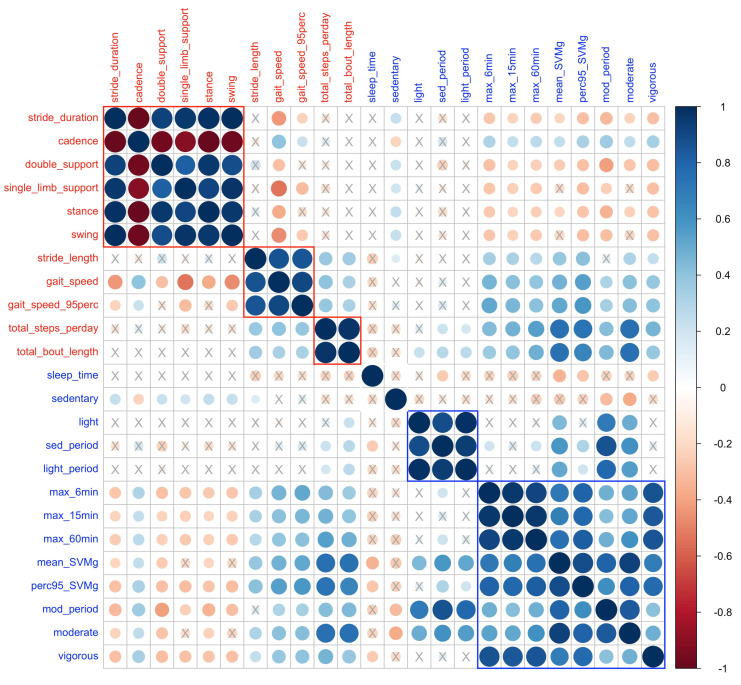
Pairwise cross-correlation matrix for gait features (colored in red) and physical activity (colored in blue) at significance level 0.1. The ‘X’ represents non-significant correlation. Three groups in gait domain (red box) and two groups in PA domain (blue box) were identified.

**Figure 4 sensors-23-08542-f004:**
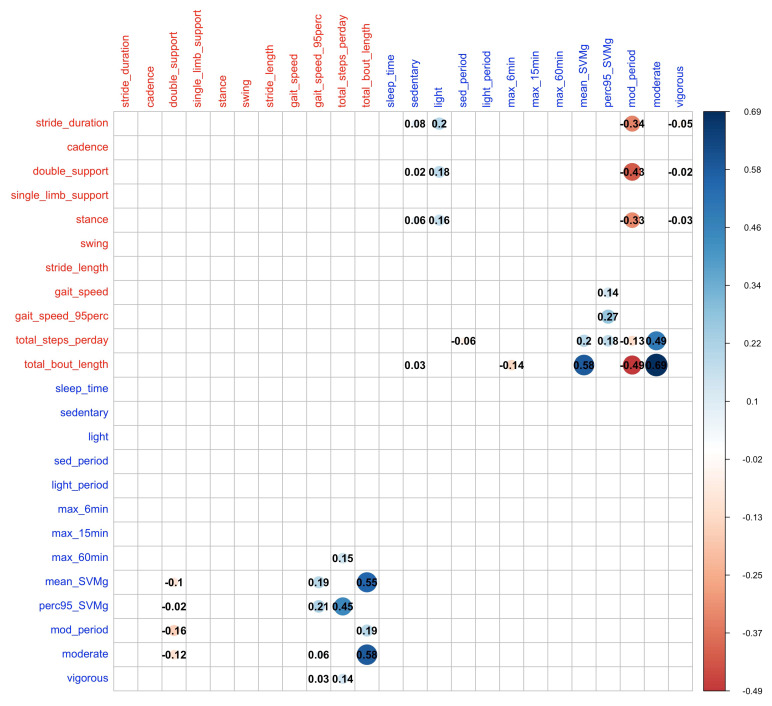
Regression coefficient (numbers in the circles) from Lasso regression, with one feature from one domain (rows) regressed on the multivariate feature domain of the other one (columns). The place was left blank if the variable was not selected in the corresponding penalized regression model. From the regression models, total bout length and total steps per day are associated with more features from the PA domain and the corresponding estimated coefficients tend to be of large value.

**Figure 5 sensors-23-08542-f005:**
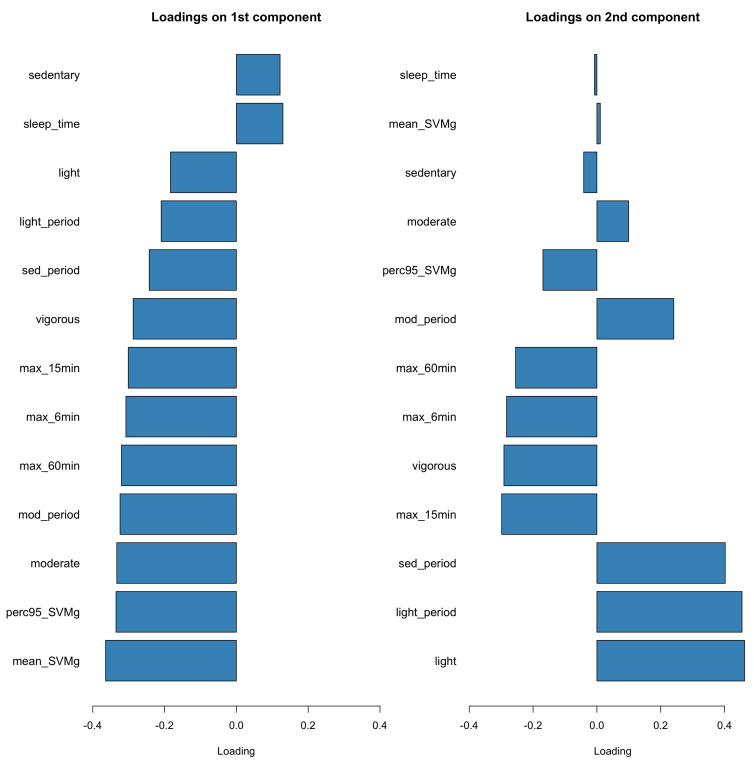
Loading vectors of the first two principal components from PCA applied to all PA features. Features related with peak activity (e.g., mean/95th percentile SVMg and moderate activity) contributed more to the first component. Features related to low-intensity activity (e.g., sedentary to light activities) contributed more to the second component.

**Figure 6 sensors-23-08542-f006:**
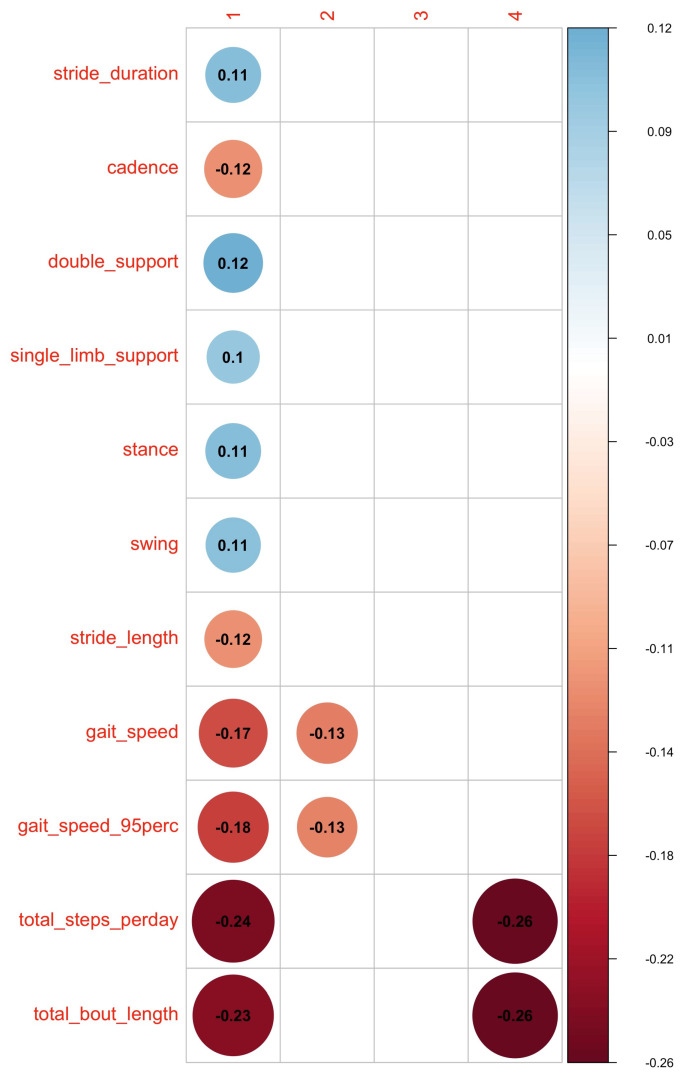
Estimated coefficients (numbers in circles) from PCR using the first four principal components from the PCA decomposition as the regression predictors. The blank box represents non-significant (*p*-value < 0.05) coefficients. PC 1 and PC 4 from the PA domain have the highest correlation with total steps and total bout length.

**Figure 7 sensors-23-08542-f007:**
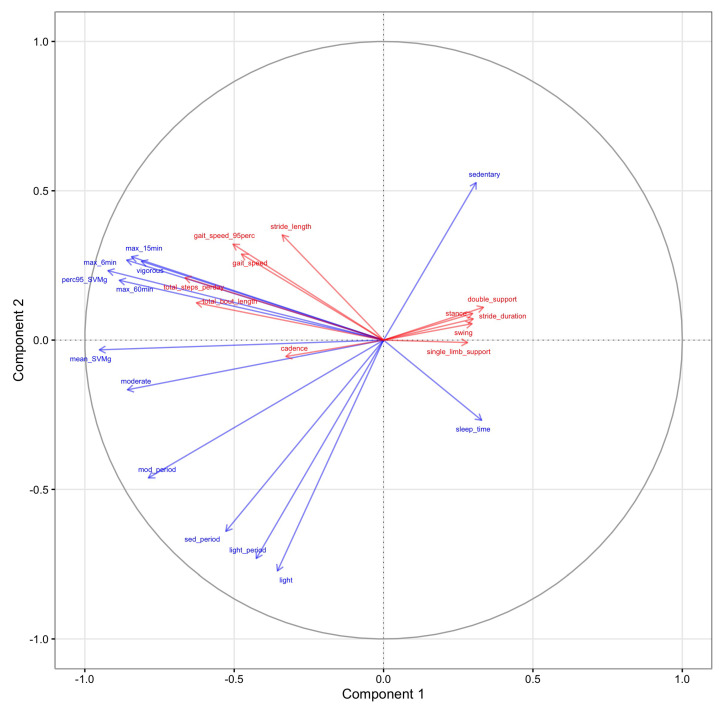
Variable correlation circle plot between gait (red) and PA (blue) domains for the first two PLS components. Two latent clusters were identified here. The first cluster included spatial and spatio-temporal features (e.g., gait speed and stride length) and gait quantity features (e.g., total steps per day and total bout length), contributing to the first component in the same direction as PA features. Another cluster included gait quality features, such as double support and stride duration. They contributed to the first component in the opposite direction of other gait features, together with sedentary activity and sleep time features from PA domain.

**Figure 8 sensors-23-08542-f008:**
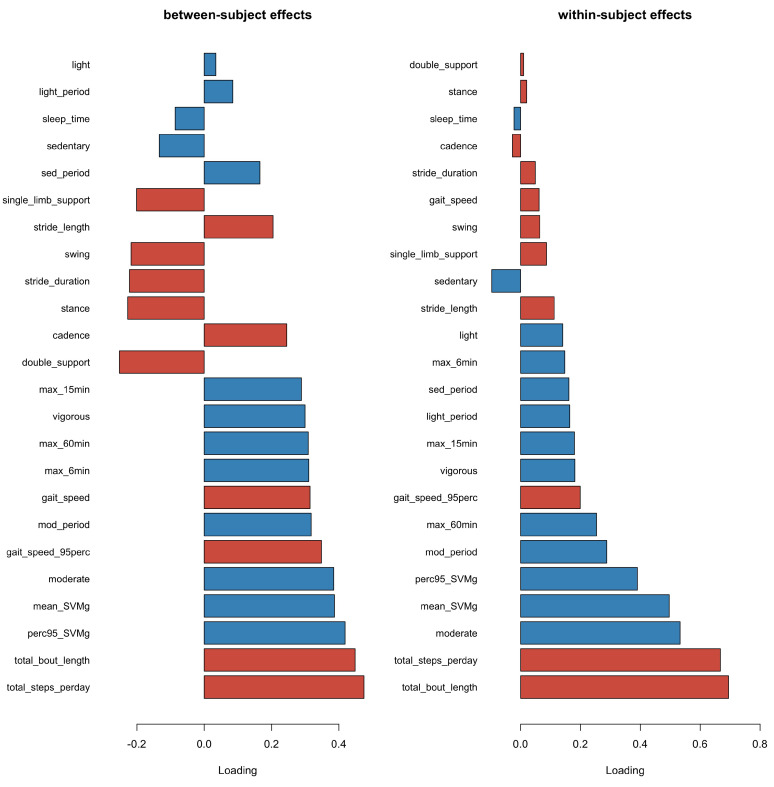
Loading plot of 1st component in multilevel PLS for between-subject (**left**) and within-subject (**right**) effects for both gait (red) and PA (blue) features. Total bout length and total steps per day from the gait domain and moderate time, mean and 95th percentile of SVMg from the PA domain had the highest contributions to the first component for both between and within-subject effects.

**Figure 9 sensors-23-08542-f009:**
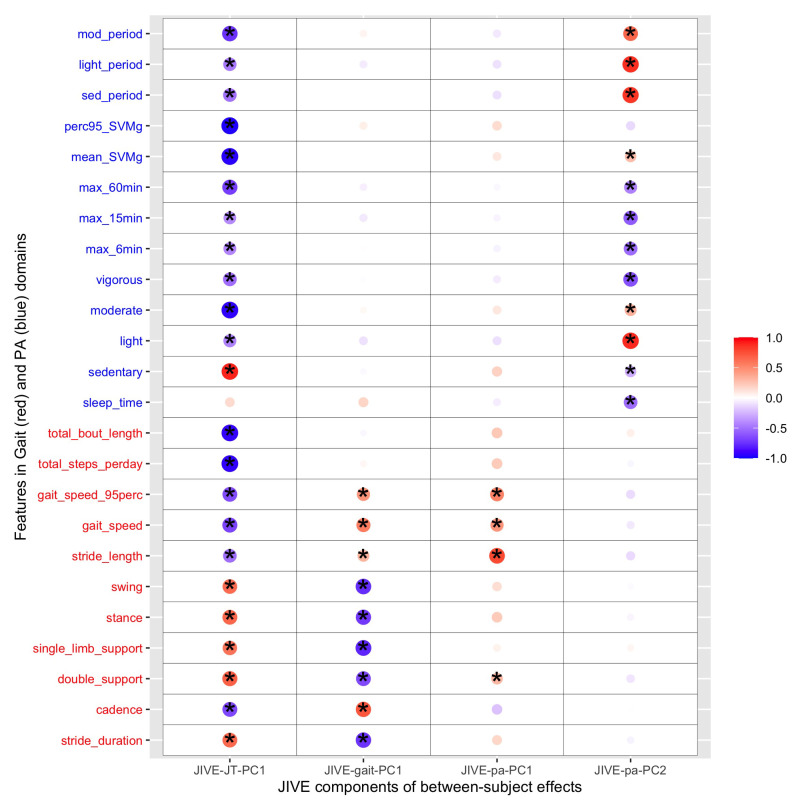
Cross-correlations between eleven gait features and thirteen PA features and the first joint score, first gait individual score, and two PA individual scores for between-subject effect from multilevel JIVE (* indicates *p*-values < 0.05). Total bout length and total steps per day from the gait domain and mean/95th percentile SVMg and moderate activity from the PA domain had the biggest contribution to the joint component in the same direction (blue dots).

**Figure 10 sensors-23-08542-f010:**
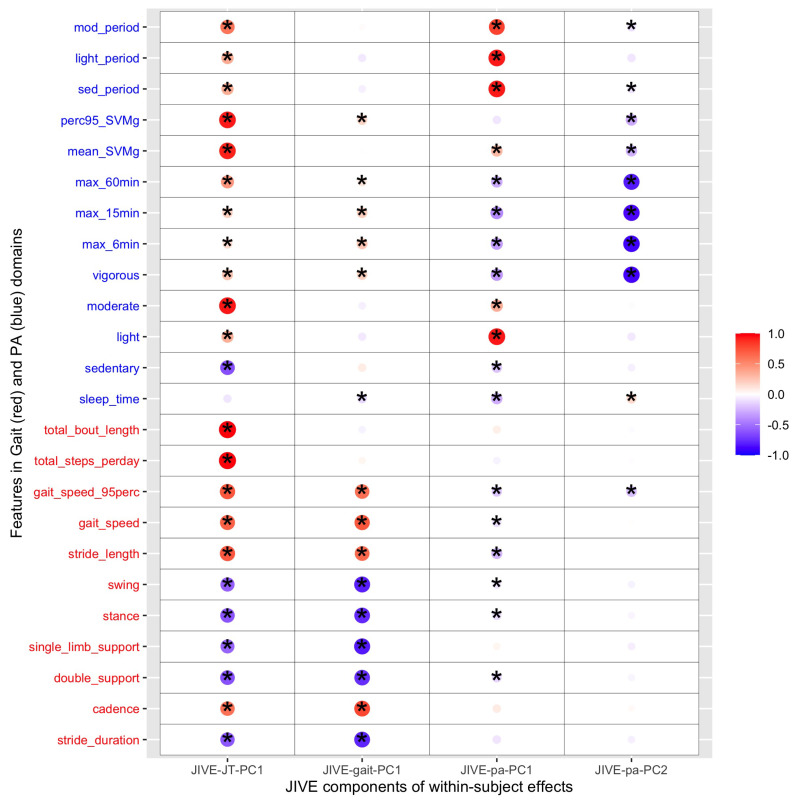
Cross-correlations between eleven gait features and thirteen PA features and the first joint score, first gait individual score, and two PA individual scores for within-subject effect from multilevel JIVE (* indicates *p*-values <0.05). Total bout length and total steps per day from the gait domain and mean/95th percentile SVMg and moderate activity from the PA domain had the biggest contribution to the joint component in the same direction (red dots).

**Figure 11 sensors-23-08542-f011:**
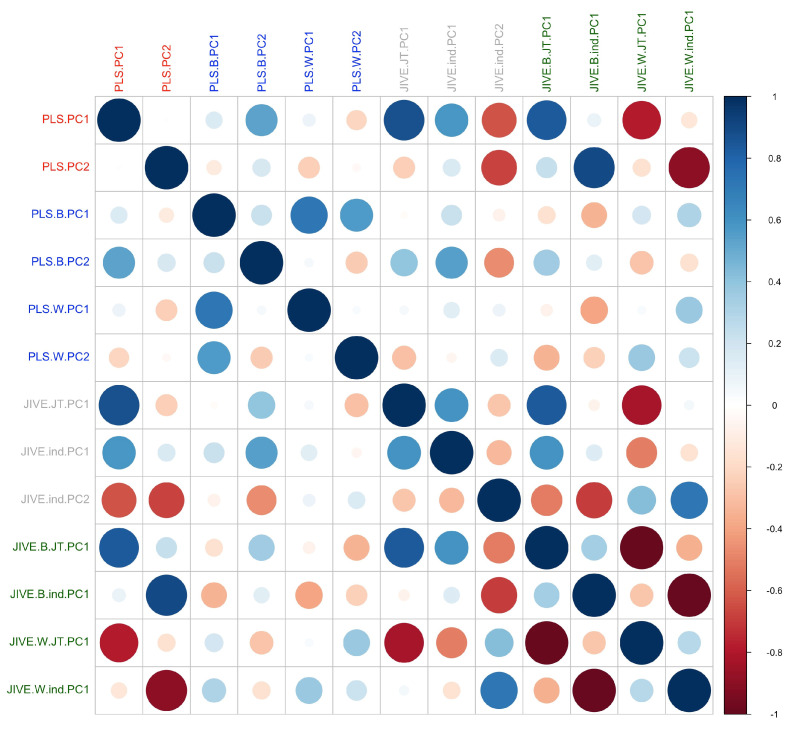
Correlations between first two components from one-level and multilevel PLS and JIVE models. First components from all models are correlated with each other, indicating the results are consistent across models. Label notations: PC1/2: 1st/2nd principal component; B.: between-subject effect; W.: within-subject effect; JT.: joint component in JIVE; ind.: individual component in JIVE.

**Table 1 sensors-23-08542-t001:** List of features and corresponding abbreviations in figures and tables.

Features in the Gait Domain	Abbreviations
total step counts per day (#)	total_step_per_day
total bout length per day (s)	total_bout_length
gait speed (m/s)	gait_speed
cadence (# steps/min)	cadence
stance time (s)	stance
swing time (s)	swing
stride duration (s)	stride_duration
double support (s)	double_support
single limb support (s)	single_limb_support
stride length (m)	stride_length
95th percentile of gait speed (m/s)	gait_speed_95perc
Features in the PA domain	Abbreviations
total duration of sedentary activity bouts (h)	sedentary
total duration of light activity bouts (h)	light
total duration of moderate activity bouts (h)	moderate
total duration of vigorous activity bouts (h)	vigorous
total daily sleep time (h)	sleep_time
total number of sedentary activity bouts (#)	sed_period
total number of light activity bouts (#)	light_period
total number of moderate activity bouts (#)	mod_period
mean SVMg	mean_SVMg
95th percentile of SVMg	perc95_SVMg
mean activity level during the most active 6-min window	max_6min
mean activity level during the most active 15-min window	max_15min
mean activity level during the most active 60-min window	max_60min

**Table 2 sensors-23-08542-t002:** Descriptive statistics (mean ± sd) of gait and PA features for all participants and younger and older cohorts separately.

Features	Overall (Mean ± Sd)	Younger (Mean ± Sd)	Older (Mean ± Sd)
total_step_per_day	7361.59 ± 3249.69	7924.53 ± 3422.65	6781.05 ± 3004.10
total_bout_length	81.71 ± 34.01	86.47 ± 36.43	76.80 ± 31.14
gait_speed	0.86 ± 0.10	0.91 ± 0.09	0.81 ± 0.09
cadence	99.21 ± 5.82	101.11 ± 5.87	97.25 ± 0.16
stance	0.76 ± 0.04	0.75 ± 0.04	0.78 ± 0.04
swing	0.47 ± 0.03	0.46 ± 0.03	0.48 ± 0.03
stride_duration	1.25 ± 0.07	1.23 ± 0.07	1.27 ± 0.07
double_support	0.29 ± 0.02	0.29 ± 0.01	0.30 ± 0.02
single_limb_support	0.49 ± 0.03	0.48 ± 0.03	0.50 ± 0.03
stride_length	1.04 ± 0.11	1.07 ± 0.11	1.00 ± 0.10
gait_speed_95perc	1.27 ± 0.17	1.35 ± 0.14	1.18 ± 0.16
sedentary	11.22 ± 1.36	10.97 ± 1.31	11.47 ± 1.40
light	1.30 ± 0.46	1.29 ± 0.31	1.32 ± 0.58
moderate	2.06 ± 0.90	2.47 ± 0.78	1.64 ± 0.83
vigorous	0.11 ± 0.14	0.16 ± 0.16	0.07 ± 0.11
sleep_time	5.79 ± 1.48	5.64 ± 1.59	5.94 ± 1.38
sed_period	60.25 ± 15.73	64.41 ± 10.10	55.96 ± 19.18
light_period	56.24 ± 17.66	57.37 ± 12.11	55.07 ± 22.13
mod_period	45.05 ± 17.06	53.10 ± 11.27	36.75 ± 18.16
mean_SVMg	131.22 ± 33.95	146.59 ± 29.82	115.36 ± 30.85
perc95_SVMg	496.51 ± 139.47	558.60 ± 117.45	432.49 ± 132.61
max_6min	921.76 ± 362.27	1100.49 ± 344.17	737.44 ± 282.13
max_15min	726.55 ± 305.06	851.77 ± 302.87	597.42 ± 252.05
max_60min	441.38 ± 170.51	502.57 ± 169.68	378.28 ± 149.03

**Table 3 sensors-23-08542-t003:** Percentages of joint and individual variation explained by each domain in JIVE.

	Joint Component	Individual Components
Explained Variation	0.30	0.56
Gait	0.58	0.47
PA	0.42	0.53

**Table 4 sensors-23-08542-t004:** Results from multilevel JIVE. 1. Between- and within-subject effects (percentages of explained variance) from MPCA decomposition. 2. Percentages of joint and individual variation explained by each domain for between- and within-subject effect in multilevel JIVE.

	**Between-Subject Effect**		**Within-Subject Effect**	
1. MPCA decomposition	0.46		0.44	
	**Joint**	**Individual**	**Joint**	**Individual**
2. JIVE decomposition	0.53	0.42	0.45	0.49
Gait	0.47	0.41	0.54	0.27
PA	0.53	0.59	0.46	0.73

## Data Availability

Upon request, and subject to review, Pfizer will provide the data that support the findings of this study. Subject to certain criteria, conditions and exceptions, Pfizer may also provide access to the related individual de-identified participant data. See https://www.pfizer.com/science/clinical-trials/data-and-results (accessed on 30 August 2023) for more information.

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
