# Peer review of "Can Gait Characteristics Be Represented by Physical Activity Measured with Wrist-Worn Accelerometers?"

_sensors, 2023, doi:10.3390/s23208542_

Round 1

Reviewer 1 Report

The manuscript is well-written, clear to understand and provides interesting insights for those doing research in the field of physical activity monitoring by means of wearable devices.

Authors could further improve the Introduction section of their work by providing the definition of Physical Activity (PA) that they take as their "reference": in fact, PA may be intended differently, also according to the type of measurements collected and sensors used. For example, for some researchers the quantification of PA is only given in terms of METs (Metabolic Equivalent of Tasks) which are not easy to estimate by means of wearable technologies. Providing the definition of PA taken as the reference one, can be useful for readers to clearly focus if this work is what they can be interested in, or not, based on "their" definition of PA (or, the definition they agree upon and work on).

Some additional comments (please, see also the annotated file):

- what type of raw signals were used? acceleration, in a digital format? or voltage values corresponding to acceleration, given in analogue format? please give details

- how do authors define the activity "level"? is it the magnitude of acceleration or what else?

Just a few typos were found, that can be easily removed.

Reviewer 2 Report

The manuscript aims to evaluate whether physical activity (PA) measures derived from a wrist-worn device can be used to represent gait measures derived from a lumbar-worn sensor. In general, the manuscript is written well. Several analytical approaches are applied to inform the results, a strength, and the results are of interest. There are, however, some major and minor points that should be addressed to improve the manuscript and its impact.

Major:

The authors argue that wrist-worn devices have been used to assess PA and lumbar devices to assess gait. While this may be the general rule, there are numerous exceptions to this, in both directions. For example, these papers and others describe how gait measures can be extracted from a wrist-worn sensor, going back at least until 2014: https://www.frontiersin.org/articles/10.3389/fphys.2021.706545/full, https://iopscience.iop.org/article/10.1088/0967-3334/35/11/2183/meta.

https://pubmed.ncbi.nlm.nih.gov/36146441/

https://pubmed.ncbi.nlm.nih.gov/31059461/

The premise and the contribution of the present work are, therefore, unclear.

Consider changing the title, abstract, and introduction to address this issue.

From that perspective, the paper could be reframed as an exploration of the relationship between PA and gait measures, although that too has been reported previously in several cohorts (albeit not as comprehensively as in the present report).

Given the previous papers that have extracted gait measures directly from wrist-worn devices, other literature, and intuition, a key conclusion of this work is rather obvious: some gait measures are reflected in PA to a large degree, but others are not (~lines 344-346).

“We then demonstrate how these methods can be leveraged in a real data application where both wrist-worn and lumbar-worn devices were deployed in the same participants simultaneously.” The authors suggest that most studies will use either a wrist-worn device or a lumbar-worn device. The rationale for this analytic exercise is not clear.

Minor:

While there is some support for the inverted pendulum model, it is not very accurate.

 In the regression analyses, were steps taken to deal with collinearities? Stride duration and cadence should, by definition, be strongly, and inversely related to each other, for example, as should single support and swing time.

 Was swing and double support expressed as % or values in time? Does that have an impact?

 “PCA was applied to the PA features and the first four components explained over 90% variation.”  Please make it clear: the variance of what?

 Please provide more information on the clinical evidence of the associations found in the statistical analysis and consider referring to relevant publications in the field to support these claims.

Tables 2 and 3 indicate that up to about 50% of the variance was explained. While 50% is high, it also suggests that ~50% was not explained. What are the implications for that?

 Figure 4: make explicit what the x-axis represents.

 Specifically, gait quantity measures, such as total steps and total bout length, and gait speed are highly correlated with PA features and can be represented by a combination of PA features.” For the reader with less expertise, please make this clearer. How highly correlated? And what combination of features? From Figure 2, it looks like none of the pairwise correlations between PA and the gait measures are especially high.

 Figure 5: what are the .. in the circles?  Is it strange that no measures were related to PCA 3?

 “However, for therapeutic areas where specific spatial and spatio-temporal features are of clinical interest, such as stride duration” Based on previous work and intuition, it seems that stride duration or cadence can be relatively accurately derived from a wrist-worn sensor, in contrast to this suggestion.

“Because gait features that are typically measured by a lumbar-worn device can be represented

by PA features, especially for those gait features related to total gait quantity and gait speed.”   Please make it clearer how and just how well this can be achieved.  

Were the results similar in the young and older adults?

As the authors note, a big open issue is what happens in the presence of pathology, gait impairment, and when study subjects use walking aids. It is unlikely that the relationships will hold in those circumstances. Please discuss these limitations.

“On the other hand, wrist-worn devices have been shown to offer better compliance when compared with lumbar-worn devices due to ease and comfort of wear resulting in lower patient burden [31,32], and therefore are preferred by patients [33].” Intuitively, the argument made here makes sense. However, it is important to keep in mind that [33] is based on a small number of subjects of a very specific patient population and that lumbar studies have also shown high compliance. A more nuanced statement may be more appropriate here; much will depend on the cohort, the type of study design, and other factors.

Were bouts from the lab sessions included in the analysis or just the daily living data? it needs to be clearer. If they were, what was the percentage of the lab gait bouts in the entire walking data?

While filtering out short and long bouts may seem reasonable, since a significant portion of daily living gait bouts are less than 10 seconds, it results in the exclusion of a substantial amount of data. This limitation should be acknowledged.  What was the rationale for the threshold for excluding long bouts?

On a related note, lumping together bouts over a very wide range may have impacted the results. Previous lumbar studies have looked at different bout lengths.

It would be beneficial to add descriptive statistics about the gait and PA measures listed in Table 1. What were the means and std (range) for each of the measures?  Maybe include this overall and for the young and older adults separately.

The multiple statistical approaches are elegant and strengthen the report. However, to have a bigger impact, please try and help the less knowledgeable reader understand what was done, why, and what was found.

In a related study of about 600 older adults that examined the association between a single test lumbar-derived gait and balance measures and PA, “three gait and balance measures were independently associated with total daily physical activity (p < .01), together accounting for approximately 16% of its variance”. https://pubmed.ncbi.nlm.nih.gov/28957994/   That finding is very different from the findings of the present study. What might explain the differences?

Table 1: “total duration/number of moderate activity bouts”: what does the / represent here and elsewhere in the table? Is this a ratio? Wouldn’t it be more informative or intuitive to switch the numerator and the denominator?

Table 1: why is vigorous reported differently from the other types of activity?

Some have suggested grouping spatial-temporal gait parameters into domains like pace, rhythm, postural control, and variability. How would that mapping influence the interpretation of the results?

Reviewer 3 Report

In this work, the authors assess whether gait characteristics, typically measured by lumbar-worn devices, could be adequately represented by physical activity (PA) features measured by wrist-worn instruments, the authors employed various statistical methods, including penalized regression, principal component regression, partial least square regression, and joint and individual variation explained (JIVE), while considering both between- and within-subject effects through multilevel models. Their findings demonstrated that PA features from wrist-worn devices can represent certain gait features, potentially reducing the number of devices required in clinical trials and improving patient comfort. Overall, the contributions of the work are significant and can be considered for publications subject to some minor yet critical corrections. 

1. The authors are suggested to highlight the limitations of existing works from the literature on the gait analysis using wrist-worn accelerometers and then tell specifically how this work differs from those works.

2. The authors should provide the schematic or factual representation of a sample subject wearing the device and participating in the gait analysis to improve the readability of the manuscript.

3. The authors mentioned in Section 3.2 that 'The gait algorithm first detects gait bouts from continuous free-living data using a gradient-boosted tree classifier and then applies wavelet transformation and inverted pendulum model to derive temporal and spatial gait features for steps in each gait bout.' However, there is not sufficient evidence in the manuscript to prove this claim.

4. Figures 8, 9, and 10 need to be explained further to understand the insights and inferences from these plots. At present, the authors have just referred to the text without adequate explanations.

Round 2

Reviewer 2 Report

The authors have addressed my concerns.